

# AMHconverter: an online tool for converting results between the different anti-Müllerian hormone assays of Roche Elecsys®, Beckman Access, and Kangrun

Huiyu Xu[1,2,3,4,*], Guoshuang Feng[5,*], Congcong Ma[1,2,3,4], Yong Han[6], Jiansuo Zhou[7], Jiatian Song[1,2,3,4], Yuan Su[1,2,3,4], Qun Zhong[1,2,3,4], Fenghua Chen[1,2,3,4], Liyan Cui[7] and Rong Li[1,2,3,4]

[1] Center for Reproductive Medicine, Department of Obstetrics and Gynecology, Peking University Third Hospital, Beijing, China
[2] Key Laboratory of Assisted Reproduction (Peking University), Ministry of Education, Peking University, Beijing, China
[3] National Clinical Research Center for Obstetrics and Gynecology, Peking University Third Hospital, Beijing, China
[4] Beijing Key Laboratory of Reproductive Endocrinology and Assisted Reproductive Technology, Beijing, China
[5] Beijing Children's Hospital, Beijing, China
[6] Hangzhou Qingguo Medical Technology Co. Ltd., Hangzhou, Zhejiang, China
[7] Department of Laboratory Medicine, Peking University Third Hospital, Beijing, China
[*] These authors contributed equally to this work.

Corresponding authors
Liyan Cui, cuiliyan@bjmu.edu.cn
Rong Li, roseli001@bjmu.edu.cn

## ABSTRACT

**Background**. The anti-Müllerian hormone (AMH) is gaining attention as a key factor in determining ovarian reserve and polycystic ovarian syndrome, and its clinical applications are becoming more widespread worldwide.

**Objective**. To identify the most accurate formula for converting AMH assay results between different platforms, so that the developed AMH converter can be used to reduce the need for multiple AMH tests at different hospitals.

**Methods**. Assuming that the Beckman Access, Kangrun, and Roche Elecsys® AMH assays fit a linear relationship from the lowest to the highest concentration (a global relationship), we used Passing–Bablok regression to determine the conversion equation between each two assays. When the relationship between two AMH assays was a local one, spline regression was used. Bland–Altman plots were drawn to check systemic bias and heterogeneity of variance across different ranges of values. The fitting effects of the models were evaluated using the squared coefficient of determination ($r^2$), adjusted $r^2$, root mean square error (RMSE), Akaike information criterion (AIC), and corrected AIC.

**Results**. The coefficient of variance for multiple controls in the Kangrun, Roche, and Beckman assays was lower than 5%, and the bias of multiple controls was lower than 7%. A global linear relationship was observed between the Kangrun and Roche assays, with the intercept being zero, for which Passing-Bablok regression was employed for data conversion between the two platforms. For the other two pairs of platforms, *i.e.*, Roche and Kangrun or Beckman and Kangrun, spline regression was applied, with the intercepts not including zero. The six corresponding formulas were developed into an online AMH converter (http://121.43.113.123:8006/).

**Conclusion**. This is the first time Passing–Bablok plus spline regression has been used to convert AMH concentrations from one assay to another. The formulas have been developed into an online tool, which makes them convenient to use in practical applications.

## INTRODUCTION

First discovered by Jost in 1947 (*Jost, 1947*), the anti-Müllerian hormone (AMH) plays an important role in sexual differentiation of male embryos, contributing to the regression of the Müllerian duct. In 1999, *Durlinger et al. (1999)* discovered that AMH inhibits primordial follicles recruitment using AMH knock-out mice, paving the way for accelerated scientific research and clinical application in reproductive science, as well as the development of AMH assay kits. Nowadays, AMH is widely recognized as an ideal marker for determining the individualized ovarian stimulation regimen of follicle-stimulating hormone (FSH) doses for *in vitro* fertilization (IVF) treatment programs (*La Marca et al., 2012*; *Xu et al., 2021a*; *Xu et al., 2023a*), and for measuring the ovarian reserve during ovarian stimulation (*Dewailly et al., 2014*; *La Marca & Volpe, 2006*). Consequently, the development of precise, efficient, and robust AMH assays is of utmost importance.

At present, a number of AMH assays are available, including automated ones that are gradually taking the place of ELISA assays (Gen II and Ultra-Sensitive AMH/MIS ELISA Kit) (*Gassner & Jung, 2014*). Among the automated assays currently on the market are Elecsys® AMH (Roche Diagnostics, Bagnolet, France), Access AMH (Beckman Coulter; USA), and Kangrun AMH (Guangzhou Kangrun, Guangzhou, China). In 2014, Roche launched Elecsys®, which utilizes monoclonal antibodies F2B 12/H and F2B 7/A (*Gassner & Jung, 2014*). This was followed by Beckman Coulter's Access platform, which also uses the same monoclonal antibodies (*Demirdjian et al., 2016*). While these two make use of the same pair of antibodoes, their respective assay development processes differ significantly. Roche's approach entails electro-chemiluminescence, biotin-crosslinked AMH antibody, ruthenium pyridine as the luminescent substance, and embedded multi-point calibration with client-side two-point adjustment. While Beckman's method involves chemiluminescence, alkaline phosphatase cross-linked AMH antibody, adamantane as the luminescent substance, and client-side multi-point calibration. A Chinese company, Kangrun, released an automated AMH assay that uses different monoclonal antibody pair, coded as 1B6 and 3D7. Reports of discrepancies in AMH values between different AMH kits has been reported (*Li et al., 2016*). This means that the results obtained by a patient in Hospital A utilizing a certain AMH kit cannot be interpreted by other hospitals utilizing a different AMH kit, thus increasing the medical burden on the patient, the complexity of

clinical practice, and hindering the clinical application of AMH-related online tool-based assessment (*Xu et al., 2021b*).

As one of the most influential predictors, we have developed various online tools based on AMH, such as for the evaluation of ovarian reserve (http://121.43.113.123:9999/) (*Xu et al., 2020b*; *Xu et al., 2020a*), the screening of polycystic ovarian syndrome (PCOS) (http://121.43.113.123:8888/) (*Xu et al., 2022*; *Xu et al., 2023b*), predicting the FSH doses for ovarian stimulation (http://121.43.113.123:8004) (*Xu et al., 2023a*), and for predicting the number of oocyte retrieved (http://121.43.113.123:8002/) (*Yong Han et al., 2022*). In this study, we used a panel of human samples to investigate the optimal formula for the transfer of serum values among different AMH tests, enabling the use of these online tools with diverse commercially available AMH assays, thus reducing the testing costs of AMH for patients while they are receiving care at different hospitals.

## MATERIALS AND METHODS

### Blood samples

For sample selection, we drew upon the previously reported distribution of AMH data from the Reproductive Centre of Peking University Third Hospital, as presented in Table 1. The samples were collected in reproductive endocrinology lab in our reproductive center. From November 2021 to February 2022, we randomly selected 300 serum AMH samples. The samples were obtained from women undergoing standard IVF at the hospital's IVF centre, and subsequently stored at $-80\ °C$. All patients provided their written informed consent for the anonymous donation of their residual serum for research purposes after the initial blood testing. Ethical approval for this study was granted by the Institutional Review Board of Peking University Third Hospital, approval number S2021543. The same sample was put through testing on three separate platforms: the Beckman and Kangrun platforms, both of which are located in the Reproductive Center laboratory, and the Roche platform, situated in the large laboratory of the hospital. All AMH assay results are reported in ng/mL.

### Assay performance

Prior to the commencement of the study, the precision and accuracy for each assay were carefully evaluated in compliance with the Clinical and Laboratory Standards Institute EP15-A3 (*Clinical and Laboratory Standards Institute (CLSI) (2014)*). Quality control sera supplied by the manufacturers (control 1 and 2 for Kangrun; PreciControl AMH 1 and 2 for Elecsys®; and controls 1, 2 and 3 for Access) were used for this purpose and each control for each assay was tested consecutively four times a day for a period of five days.

For each sample, two tubes of serum were frozen; one of which was thawed and tested on the Roche platform, while the other was thawed and tested on the same day using the Kangrun and Beckman platforms. When results exceeded the upper measurement range of any of the methods, the serum samples were automatically diluted with reagents provided by the manufactures and analysed again. The Roche Elecsys® AMH immunoassay was run on the Roche Cobas e801, the Beckman Access AMH assay on the Beckman Coulter

DxI 800, and the Kangrun AMH assay on the Kaeser6600. Respectively, these three AMH assays had linear detection ranges of 0.02–24 ng/ml, 0.01–23 ng/ml, and 0.06–18 ng/ml.

## Statistical analysis

Assuming a global relationship between the Beckman, Kangrun, and Roche assays, which means that there is an algorithm that fit the full detection range of AMH concentration, a Passing–Bablok regression was employed in order to determine the conversion equation between the respective pairs. This nonparametric method was used to calculate the parameters 'a' and 'b' for the linear equation $y = a + bx$. The 95% confidence interval of the intercept 'a' and the slope 'b' were then checked to assess for any systematic or proportional differences between the two AMH assays.

In case of a local relationship, a spline regression was performed to obtain the conversion formula. This piecewise polynomial function is smoothly connected at the nodes, and, as per the relationship established between the independent and dependent variables in this study, a linear spline regression was used, resulting in a linear regression with smooth connections at the nodes. Quality controls provided with the AMH kits were used to evaluate the detection values, with deviations from the target value expressed as bias. The acceptable criteria for the total error, bias, and coefficient of variance were set at $\pm 25\%$, $\pm 12\%$, and $\leq 8\%$, respectively.

To evaluate systemic bias and heterogeneity of variance across different ranges of values, a Bland–Altman plot was created. Model fitting effects were then assessed using the squared coefficient of determination ($r^2$), adjusted $r^2$, root mean square error (RMSE), Akaike information criterion (AIC), and corrected AIC (AICc). The higher the $r^2$ and adjusted $r^2$, the better the model performance, while the lower the RMSE, AIC, and AICc, the better the fitting effect. All statistical analyses were conducted using JMP PRO *v.* 16.0 (Cary. NC, USA).

## RESULTS

The results for precision and accuracy of each AMH assay are shown in Table 2. The CVs for compared sets of analytic and evaluated AMH values were satisfactory. For each assay, the analysed CV value was $\leq 5\%$, with bias of $\pm 7\%$.

### Passing–Bablok regression among the three AMH assays

Assuming that the Roche and Kangrun AMH assays fit a global relationship, Passing–Bablok regression was utilized to construct a linear relationship between the two assays. Figure 1A shows the fitting results. In Fig. 1A, the dashed line illustrates the ideal perfect agreement between the two AMH assays, and the blue line is the Passing–Bablok regression. As seen in Fig. 1A, the blue line diverges from the dotted line; this disparity needs to be statistically assessed.

The Bland–Altman plot was used to explore the agreement between two AMH assays. As shown in Fig. 1B, the horizontal axis denotes the mean of Kangrun and Roche measurements, while the vertical axis reflects the difference between Kangrun and Roche measurements. The grey solid line in Fig. 1B is the reference line, denoting 0 value

**Table 1  Sample collection criteria according to previous AMH distribution at our center.**

| Range | AMH (ng/ml) | Number of samples |
|---|---|---|
| | <0.2 | 30 |
| | 0.2-<0.4 | 30 |
| Low | 0.4-<0.6 | 30 |
| | 0.6-<0.8 | 30 |
| | 0.8-<1.0 | 30 |
| | 1.0-<1.5 | 30 |
| Mid | 1.5-<2.5 | 30 |
| | 2.5-<3.5 | 30 |
| High | 3.5-<15.5 | 30 |
| | ≥15.5 | 30 |

**Table 2  Precision and accuracy of the three AMH assays.**

| Assay | Control Sample | Batch lot No. | Provided target value (ng/ml) | Range | Average (ng/ml) | SD | CV (%) | Bias (%) |
|---|---|---|---|---|---|---|---|---|
| Roche Elecsys | Control 1 | 54302201 | 1.22 | 1.13–1.19 | 1.17 | 0.02 | 1.49 | −4.18 |
| | Control 2 | | 5.92 | 5.89–6.02 | 5.89 | 0.1 | 1.78 | −0.45 |
| Access Beckman | Control 1 | 189204 | 1.01 | 0.86–1.02 | 0.95 | 0.04 | 4.51 | −6.24 |
| | Control 2 | | 5.13 | 4.66–5.41 | 5.02 | 0.23 | 4.51 | −2.19 |
| | Control 3 | | 15.2 | 13.84–15.25 | 14.37 | 0.34 | 2.38 | −5.44 |
| Kangrun AMH | Control 1 | 20210806 | 2.39 | 2.19–2.51 | 2.37 | 0.09 | 3.67 | −0.92 |
| | Control 2 | | 5.8 | 5.45–6.06 | 5.76 | 0.21 | 3.59 | −0.69 |

**Notes.**
AMH, anti- Müllerian hormone; SD, standard deviation; CV, coefficient of variance.

(representing the mean difference value of 0 between two AMH platforms); the red solid line and red dotted line indicate the mean and 95% CI of Kangrun minus Roche AMH measurements. The grey dotted line stands for the 95% distribution range of the difference (mean ± 1.96× standard deviation), commonly known as the limit of agreement. Generally speaking, the closer the red solid line to the reference line, the more consistent the two measurements. If 95% of the points are located within the dashed grey line and follow a normal distribution around the red solid line, then it can be concluded that these two AMH assays are consistent. In this study, 8.5% of observations exceeded the grey line. The points exceeding the consistency limit showed that Kangrun measurements were greater than Roche measurements, and the distribution was evidently non-normal, particularly the high-value part. Consequently, it cannot be assumed that these two AMH assays are consistent.

Figure 1C shows a mountain plot, with the abscissa indicating the value of Kangrun minus Roche and the ordinate indicating the percentile of the cumulative distribution. When the cumulative distribution probability is below 50%, this is the cumulative distribution probability, whereas when it is above 50%, it is 1 −cumulative distribution probability.

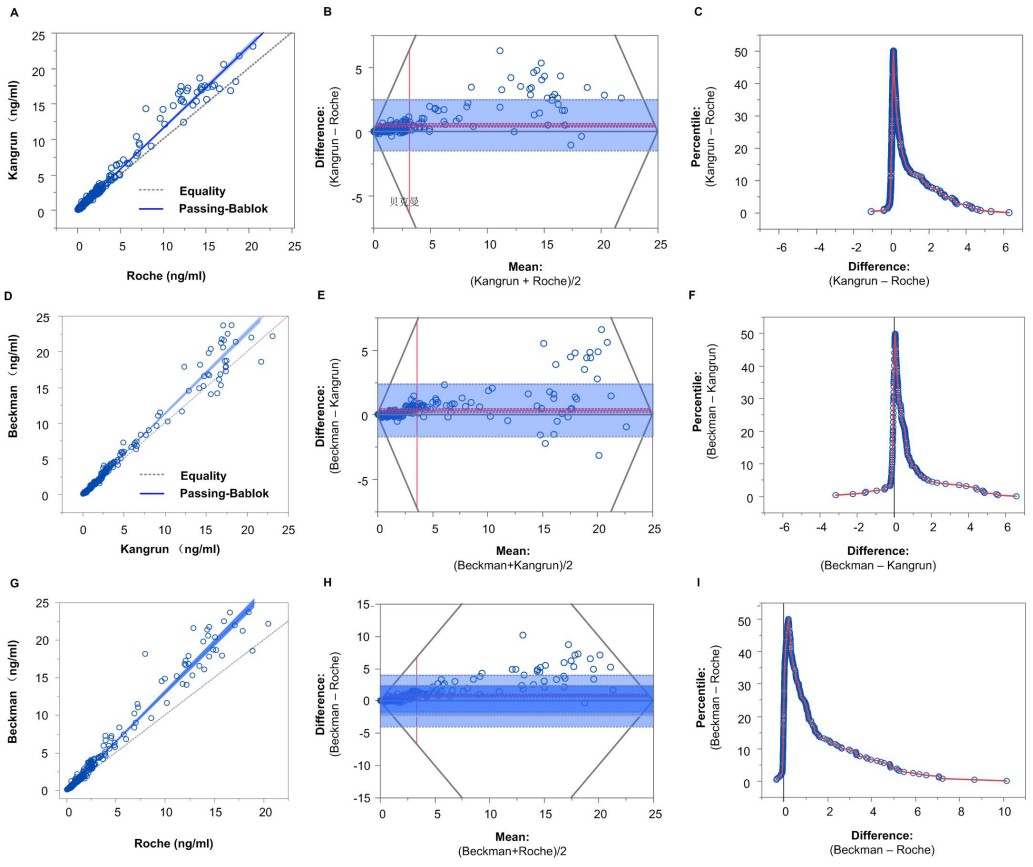

**Figure 1** **(A–I) Passing–Bablok regression between Roche and Kangrun AMH assays.** (A, D, G) Linear relationship between the two assays. (B, E, H) Bland–Altman plot, showing the systemic bias and heterogeneity of variance of the two assays. (C, F, I) Mountain plot, showing the distribution of Kangrun minus Roche. When the cumulative distribution probability is less than 50%, this is the cumulative distribution probability, and when it is greater than 50%, this is 1-cumulative distribution probability. If the two AMH assays are consistent, they should be centred at 0 and distributed symmetrically.

If the two AMH assays are consistent, they should centered around 0 and display a symmetrical distribution. As can be seen in Fig. 1C, 'Kangrun −Roche' shows a right-skewed tail, indicating that the difference between Kangrun and Roche is mostly greater than 0, and the right tail is long, suggesting that some points of 'Kangrun −Roche' are very high. Nevertheless, when the cumulative distribution was less than 50%, there was only a small difference between the two AMH assays.

The intercept converting AMH from Roche to Kangrun was −0.0124, with its 95% CI including 0, indicating that there is no systematic difference in AMH values between Kangrun and Roche assays. The slope was 1.1517, and the 95% CI did not include 1, suggesting a possible proportional difference between the two AMH assays. Using Passing–Bablok regression, we performed other pairwise transformations among the three AMH assays. The results of the pairwise transformation between the three AMH assays are

**Table 3 Conversion of AMH assays between Roche, Beckman and Kangrun using Passing-Bablok regression.**

|  | Intercept (95% CI) | Slope (95% CI) |
|---|---|---|
| Kangrun to Beckman | −0.1027 (−0.1264, −0.0786) | 1.1386 (1.1194, 1.1607) |
| Kangrun to Roche | 0.0108 (−0.0035, 0.0303) | 0.8683 (0.85, 0.8844) |
| Beckman to Roche | 0.0868 (0.07, 0.1129) | 0.7632 (0.7429, 0.7789) |
| Roche to Beckman | −0.1138 (−0.1515, −0.0899) | 1.3103 (1.2840, 1.3462) |
| Roche to Kangrun | −0.0124 (−0.0356, 0.00393) | 1.1517 (1.1307, 1.1765) |
| Beckman to Kangrun | 0.0902 (0.0702, 0.1089) | 0.8783 (0.8615, 0.8933) |

Notes.
AMH, anti-Müllerian hormone; CI, confidence interval.

**Table 4 Selective display of six samples of AMH assay conversions using Passing-Bablok regression.**

| Samples | Original values | | | Converted values | | |
|---|---|---|---|---|---|---|
|  | Beckman | Roche | Kangrun | Kangrun to Beckman | Kangrun to Roche | Beckman to Roche |
| 1 | 0.08 | 0.08 | 0.06 | −0.03 | 0.06 | 0.15 |
| 2 | 0.43 | 0.39 | 0.46 | 0.42 | 0.41 | 0.41 |
| 3 | 1.21 | 1.04 | 1.08 | 1.13 | 0.95 | 1.01 |
| 4 | 3.02 | 2.23 | 2.38 | 2.61 | 2.08 | 2.39 |
| 5 | 10.21 | 6.90 | 10.35 | 11.68 | 9.00 | 7.88 |
| 6 | 23.70 | 18.50 | 18.12 | 20.53 | 15.74 | 18.17 |

depicted in Figs. 1D–Figs. 1I and summarized in Table 3, while the converted data of the 6 selected samples can be found in Table 4 (with details in Table S1).

## Spline regression

Passing–Bablok regression supposes that the correlation between two AMH assays is not a local relationship. However, as Table 3 demonstrates, in the four conversions with 95% CI of intercepts that do not include 0, the fitting effect of the low-value part appears to be unsatisfactory, implying a potential local relationship between each pair. Therefore, spline regression was conducted.

Taking the Roche to Beckman AMH assay for an example. The value obtained from Roche measurement was taken the independent variable $x$, while the value obtained from Beckman measurement was taken as the dependent variable $y$. Based on the relationship between the independent and the dependent variable in our data, linear spline regression was used to construct a linear regression with smooth connections at the nodes. A selection of nodes was based on the principal that the smaller the variation, the better the formula, as indicated by the scatter plot and the actual distribution of AMH values in our data. Therefore, two nodes of '1 ng/mL' and '9 ng/mL' were chosen. Table 5 shows a comparison of the fitting effects between spline regression and Passing–Bablok regression. It was evident that the RMSE, AIC, and AICc of spline regression were lower than those in Passing–Bablok regression, while the $r^2$ and adjusted $r^2$ were higher than those in Passing–Bablok regression.

**Table 5  Comparison of spline regression and Passing–Bablok regression when converting from Roche to Beckman AMH assay.**

|  | Spline regression | Passing-Bablok regression |
|---|---|---|
| RMSE | 0.858 | 0.943 |
| $R^2$ | 0.977 | 0.972 |
| adjusted $R^2$ | 0.977 | 0.972 |
| AIC | 207.138 | 260.130 |
| AICc | 207.349 | 260.214 |

Notes.

AMH, anti-Müllerian hormone; RMSE, root mean square error; AIC, Akaike information criterion; AICc, corrected AIC; AMH, anti-Müllerian hormone.

Table 6 reveals the transformed values of AMH platforms with intercepts not containing 0, including the results of spline regression and Passing–Bablok regression; all converted data are presented in Table S2. It is evident that, while the overall difference between spline regression and Passing–Bablok regression is small, the lower-value is evidently better when using spline regression. For the mutual conversion between Roche and Kangrun with the intercept including 0, both spline regression and Passing–Bablok regression have satisfactory fitting outcomes in the low-value part. The formula based on Passing–Bablok regression for the Roche–Kangrun conversion is preferred due to its simplicity. Table S2 shows the converted values using spline regression and Passing–Bablok regression among the three platforms. We have also set up an online tool for AMH assay conversions (http://121.43.113.123:8006/). Therefore, for conversions with intercepts not including 0, the spline regression formula was adopted, and for conversions between Roche and Kangrun with intercepts containing 0, the Passing–Bablok regression formula was utilized.

## DISCUSSION

Recently, AMH has been widely recognized as a key factor in regulating ovarian reserve and PCOS (*Xu et al., 2021a; Teede et al., 2019; Tata et al., 2018; Durlinger et al., 1999*). However, the results of different AMH assessments vary significantly (*Li et al., 2016; Nelson et al., 2015; van Helden & Weiskirchen, 2015; Ferguson et al., 2018; Ferguson et al., 2020*), proving to be a barrier to the utilization AMH-related artificial intelligence tools (*Xu et al., 2021b*). This has caused multiple AMH tests to be carried out in various assisted reproductive technology (ART) centers or hospitals. In this study, we are the first to suggest the use of Passing–Bablok or spline regression to convert AMH concentration between assays. To put it simply, when the 95% confidence interval of the intercept of Passing-Bablok regression contains '0', Passing-Bablok regression should be applied,, such as the conversion between Roche and Kangrun AMH. On the other hand, if the 95% confidence interval of the intercept of Passing-Bablok regression does not include '0', then spline regression should be conducted, for example, AMH conversion between Beckmann and Kangrun or between Beckmann and Roche. Moreover, this formula has been developed into an online tool (http://121.43.113.123:8006/), which is convenient in practical application.

**Table 6  Data conversions using spline regression when the 95% CI of the intercept does not include '0' using Passing-Bablok regression.**

| Samples | Original data | | | Converted values | | | |
| | Beckman | Roche | Kangrun | Kangrun to Beckman | | Roche to Beckman | |
| | | | | Spline regression | Passing-Bablok regression | Spline regression | Passing-Bablok regression |
|---|---|---|---|---|---|---|---|
| 1 | 0.08 | 0.05 | 0.08 | 0.11 | −0.01 | 0.06 | −0.05 |
| 2 | 0.39 | 0.38 | 0.36 | 0.31 | 0.31 | 0.41 | 0.39 |
| 3 | 1.21 | 1.04 | 1.08 | 1.11 | 1.13 | 1.12 | 1.25 |
| 4 | 2.67 | 2.33 | 2.52 | 2.83 | 2.77 | 3.10 | 2.94 |
| 5 | 11.47 | 7.21 | 9.43 | 11.09 | 10.63 | 10.59 | 9.33 |
| 6 | 22.50 | 16.00 | 17.64 | 19.17 | 19.98 | 20.15 | 20.85 |

**Notes.**

Key, CI, confidential interval.

We know that if there is a significant large difference with fresh and freeze-thaw sample, our algorithm using freeze-thaw samples will not work in fresh samples. Thus, we conducted a comparison of the variation between fresh samples and freeze-thaw samples using Kangrun platform. Specifically, a total of 8 samples were included, stored at −20 °C, and freeze and thaw for 7 times. Repeated measures ANOVA was used to analyze the data. The results showed that the freeze-thaw had no impact on serum AMH concentration (File S1). We also adopted repeated measures ANOVA method for the data provided by Roche, and the results also showed that freeze-thaw had no effect on serum AMH concentration (File S1). The AMH assays of Beckman and Roche share the same antibody pair, and we speculate that the effect of freeze-thaw treatment maybe the same using Beckman AMH assay. In addition, previous studies have shown that the variations between fresh and freeze-thaw samples were very small, with only a decrease of about 3.9% (Roche) and 4.1% (Beckmann) at −80 °C (*Li et al., 2016*). It is uncertain whether this small difference is caused by experimental error of different tests. In conclusion, there may be no variation or at least small variation between fresh and freeze-thaw samples, so we chose the freeze-thaw samples in our study.

Our idea in this study is to first apply Passing–Bablok regression to detect whether there is a systematic difference, *i.e.,* to determine whether the 95% CI of the intercept includes 0. If the intercept includes 0, Passing–Bablok regression can be utilized to carry out assay conversion between the two platforms. On the other hand, if there is a systematic difference, spline regression is applied. The AIC method can be applied to select the number of nodes during spline regression, with the smaller the AIC indicating lesser variation. The specific values of these nodes should be refined and slightly adjusted based on the calculated converted values and their original values. We believe that this platform conversion method can be applied for various assay conversions, not only AMH assays. In fact, we have conducted other assay conversions, such as for human chorionic gonadotropin and oestradiol. For those with a wide calibration range and an intercept using Passing–Bablok regression much larger than 0, spline regression is always required. We have previously performed conversion from Kangrun to Anshlab AMH assay using the

'Passing–Bablok or spline regression' method. The formula has been incorporated into the Kangrun instrument in our laboratory; thus, the AMH values from Kangrun can be easily converted into Anshlab AMH values, which is benificial for clinicians who are familiar with our previous Anshlab AMH assay. The conversion is also advantageous for the use of Anshlab AMH data-based online tools (*Xu et al., 2023a*; *Xu et al., 2020b*; *Xu et al., 2020a*; *Xu et al., 2022*; *Xu et al., 2023b*; *Yong Han et al., 2022*; *Han et al., 2022*).

An international standard (IS) for AMH, coded 16/190 and containing lyophilised recombinant human AMH, was established in a World Health Organization collaborative study (*Ferguson et al., 2020*). In this study, sixteen immunoassay platforms that met the validity criteria showed a geometric mean estimate of 511 ng/ampoule (95% CI [426–612] CV 42%) and a robust geometric mean of 489 ng/ampoule. Despite the significance of having an IS for AMH measurement, there is wide variation in the concentration for an AMH IS using different AMH assays, with estimates ranging from 282 ng/ampoule to 1,157 ng/ampoule. Therefore, it is not surprising that discrepancies have been observed among different commercially available assays (*Li et al., 2016*; *Nelson et al., 2015*; *van Helden & Weiskirchen, 2015*). A possible reason for the observed inter-assay variability may be the differences in antibody affinity, parameters of the reaction system, or assay calibration (*Ferguson et al., 2020*).

In our study, we discovered the following. First, there was no systematic difference between Roche and Kangrun, the 95% CI of the intercepts included 0, and only proportional disparities were present. One possible cause may be that the Kangrun AMH assay uses an acridine ester chemiluminescence system, and Roche uses a ruthenium pyridine electrochemiluminescence system; acridine ester and ruthenium pyridine are similar small molecules. Second, the concentration of AMH among the three AMH assays did not differ much in the low-value part (as shown in Tables S1 and S2). We suppose that the use of an AMH IS may contribute to the small differences in the low-concentration part.

### Limitations

Although we proposed novel concepts for platform conversion in this study, this research has some restrictions. First, all our sample tests were completed within 2–3 days. Reagents with the same calibration and the same batch number were utilized; yet, when reagents with disparate calibrations and different batch numbers are taken into consideration, bias may occur. In the future, the parameter estimates of the formula should be further defined using bigger validation datasets, thus making the formula have a more extensive applicability. Moreover, we also used different quality control products provided within each individual AMH kit. If distinct commercially available AMH assays used the same third-party AMH quality control products, this could aid in the comparability among the different AMH assays.

## CONCLUSION

Three commercially available AMH assays in our study exhibited excellent performance in terms of accuracy and precision. Moreover, the conversion of the three assays in our study allows for one AMH assay to be converted to another without additional testing, this

facilitating wider utilization of AMH-related tools, as well as helping to reduce the cost for patients undergoing ART when visiting different clinics.

## ACKNOWLEDGEMENTS

We thank Liwen Bianji (Edanz) for editing the English text of a draft of this manuscript.

### Funding

This study was supported by the National Natural Science Foundation of China for Distinguished Young Scholars (grant no. 81925013); the Innovation & Transfer Fund of Peking University Third Hospital (grant nos. BYSYZHZB2020102, BYSYZHKC2021104); Major National R&D Projects of China (grant no. 2017ZX09304012-012); the National Natural Science Foundation of China (grant no. 81771650); the Capital Health Research and Development of Special Project (grant no. 2018-1-4091); and the Beijing-Tianjin-Hebei Basic Research Cooperation Project (grant no. 19JCZDJC65000). The funders had no role in study design, data collection and analysis, decision to publish, or preparation of the manuscript.

### Grant Disclosures

The following grant information was disclosed by the authors:
National Natural Science Foundation of China for Distinguished Young Scholars: 81925013.
Innovation & Transfer Fund of Peking University Third Hospital: BYSYZHZB2020102, BYSYZHKC2021104.
Major National R&D Projects of China: 2017ZX09304012-012.
National Natural Science Foundation of China: 81771650.
Capital Health Research and Development of Special Project: 2018-1-4091.
Beijing-Tianjin-Hebei Basic Research Cooperation Project: 19JCZDJC65000.

### Competing Interests

Yong Han is employed by Hangzhou Qingguo Medical Technology Co. Ltd.

### Author Contributions

- Huiyu Xu conceived and designed the experiments, performed the experiments, prepared figures and/or tables, authored or reviewed drafts of the article, and approved the final draft.
- Guoshuang Feng conceived and designed the experiments, performed the experiments, analyzed the data, prepared figures and/or tables, authored or reviewed drafts of the article, and approved the final draft.
- Congcong Ma conceived and designed the experiments, performed the experiments, analyzed the data, authored or reviewed drafts of the article, and approved the final draft.

- Yong Han performed the experiments, analyzed the data, authored or reviewed drafts of the article, and approved the final draft.
- Jiansuo Zhou performed the experiments, authored or reviewed drafts of the article, and approved the final draft.
- Jiatian Song performed the experiments, authored or reviewed drafts of the article, and approved the final draft.
- Yuan Su analyzed the data, authored or reviewed drafts of the article, and approved the final draft.
- Qun Zhong analyzed the data, prepared figures and/or tables, authored or reviewed drafts of the article, and approved the final draft.
- Fenghua Chen analyzed the data, authored or reviewed drafts of the article, and approved the final draft.
- Liyan Cui performed the experiments, authored or reviewed drafts of the article, and approved the final draft.
- Rong Li conceived and designed the experiments, authored or reviewed drafts of the article, and approved the final draft.

## Human Ethics

The following information was supplied relating to ethical approvals (i.e., approving body and any reference numbers):

Ethical approval for this study was granted by the Institutional Review Board of Peking University Third Hospital, approval number S2021543.

## Data Availability

The raw measurements are available in the Supplemental Files.

## Supplemental Information

Supplemental information for this article can be found online at http://dx.doi.org/10.7717/peerj.15301#supplemental-information.

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
