# Peer review of "AMHconverter: an online tool for converting results between the different anti-Müllerian hormone assays of Roche Elecsys®, Beckman Access, and Kangrun"

_PeerJ, doi:10.7717/peerj.15301_

## Round 0.1 · original submission · Major Revisions

While both reviewers and I recognise some potential interest, there are major issues that must be addressed before we can consider this as suitable for publication in PeerJ.

These include how the different assays behave using fresh versus frozen serum, and reagent lot variability. These variables need to be systematically addressed.

I would also agree that Bland-Altman plots are an essential requirement for comparisons of this type.

Should you be willing to address these and the other issues raised by both reviewers with new experiments, then I will take a second look.

Reviewer 1 ·

Basic reporting

No comment

Experimental design

No comment

Validity of the findings

I am not sure this is something useful and relevant for improving patient healthcare. The three methods are undoubtedly very similar to each other in performance (same international standard, partially same antibodies) and not much knowledge is obtained by using such converter. This approach might be more interesting for some parameters to obtain a cutoff when using a different method. Also it would be interesting to have more information concerning linearity and low detection limit for all three methods in order to better evaluate the range of measurements for each.
Nevertheless, I am not expert in this kind of approach and I would recommend the advice of more qualified reviewer.

Reviewer 2 ·

Basic reporting

This manuscript compares 3 different AMH assays with the intention of creating a converter tool for patient results.

The Introduction should elaborate further on the assay differences. It is not sufficient to state only that “the main difference between them is in their assay development” (line 73). Salient points about the calibrator used and biotin interference can be moved from the Discussion.

The meaning of “fit a global relationship” is not clear. Please explain.

Experimental design

I commend the authors on conducting sample measurements for all 3 methods at the same time. The only question this raises is whether the relationship of the assay data would differ if fresh serum had been used? Do you have frozen serum stability data for AMH on each assay method?

The study limitation of reagent lot variability is a significant concern and could easily obviate the usefulness of the converter tool. Multiple lots must be tested for each assay.

State specifically how each assay is calibrated in the Methods please. Also provide the AMH epitopes recognized by the different antibodies and the lower limits of detection for each assay.

Bland Altman graphs would be helpful for all of the assay comparisons. Table 6 could be eliminated in this case.

The amount of data in Tables 4 and 7 should be reduced. Showing two high, medium, and low data points would suffice. It is also not necessary to show the relationships between assays in both directions. Select one assay as the reference and compare the two others in one direction only.

Validity of the findings

The aim and study design are clear but a simple and general statement of the relationship between the assays is needed.

---

## Round 0.2 · Minor Revisions

A few issues remain to be addressed which are elaborated on in the review below. Please can you pay particular attention to the clarity of writing throughout as it is important your message is clearly conveyed to the readers.

Reviewer 2 ·

Basic reporting

English grammar requires some correction. Some typographical errors also occur (simplify).

The most common terminology to use for method comparison studies are correlation or no correlation (with r and p values provided) and proportional or non-proportional differences. The term "full scale relationship" is not clear and should not be used.

Rewrite the new sentence regarding the lower limit of detection for each assay.

Experimental design

The abstract should specify which assay conversions were done using spline regression (ie Beckman versus Roche, etc.)

Specify where (outside laboratories) testing was done for the Beckman and Roche methods.

Validity of the findings

The conversion formulas developed can only be applied on a limited basis.

Additional comments

The additional sample stability data can be added as a supplemental file.

---

## Round 0.3 · accepted · Accept

Thank you for your efforts at improving the readability of the study. I am happy to accept it now.